# Analysing the Large-scale Debris Flow Event in July 2022 in Horlachtal, Austria Using Remote Sensing and Measurement Data

Jakob Rom [1,*], Florian Haas [1], Florentin Hofmeister [2], Fabian Fleischer [1], Moritz Altmann [1], Madlene Pfeiffer [3], Tobias Heckmann [1] and Michael Becht [1]

[1]   Physical Geography, Catholic University of Eichstätt-Ingolstadt, 85072 Eichstätt, Germany
[2]   Hydrology and River Basin Management, Technical University of Munich, 80333 Munich, Germany
[3]   Institute of Geography, University of Bremen, 28359 Bremen, Germany
*   Correspondence: jrom@ku.de

**Abstract:** High-quality in situ measurements are essential for hazard assessment of debris flow events. However, precise data on debris flow triggering thresholds, accumulation volumes and spatial characteristics of large-scale events on catchment scale are scarce due to the rare occurrence of debris flows and the challenges of acquiring accurate data for a larger area. In this study, we present quantitative analyses of a single extreme debris flow event in the Horlachtal, Austria, triggered by local high-intensity short-duration precipitation events on 20 and 23 July 2022. Pre- and post-event airborne LiDAR (light detection and ranging) data with a high spatial resolution reveal that 156 different debris flow processes were initiated during these events, with accumulation volumes of up to approximately 40,000 $m^3$. The calculated debris flow deposition volumes also show a power-law relationship with the total amount of rainfall in the respective debris flow catchments. The spatial appearance of the debris flows shows a concentration of processes in a particular area rather than a uniform distribution, suggesting a local nature of the triggering event. This is further supported by the measurements from three meteorological stations and four discharge gauges within the study area. The gridded area-wide INCA (Integrated Nowcasting through Comprehensive Analysis) rainfall data further point to a local convective event on 20 July 2022, with a maximum rainfall intensity of 44 mm/h.

**Keywords:** debris flows; precipitation event; Austrian Alps; Horlachtal; LiDAR

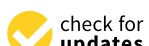



## 1. Introduction

In the summer of 2022, several thunderstorms with extreme rainfall intensities triggered a series of debris flows within the wider Stubai Alps region of Tyrol, Austria. These caused extensive damage to infrastructure and buildings, and even resulted in one fatality.

Especially during such heavy precipitation events, debris flows occur as a natural hazard in all mountainous regions around the world [1]. Due to the high-risk potential of debris flows, it is necessary to understand their triggering mechanisms as well as their flow dynamics. Thus, it is of great importance to acquire in situ field data of debris flow systems [2] especially after large-scale events. Because of the rare occurrence of debris flow events, there is a lack of direct observations of the process and its consequences. Permanently installed debris flow monitoring systems have been established in some parts of the Alps, such as Italy [3–5], France [6] and Switzerland [7,8], to gain observational data. In such systems, a wide range of precise instruments provide accurate data on e.g., rainfall conditions, flow dynamics and velocities [2]. However, these installations focus mainly on channelised debris flows at medium-to-low altitudes and provide detailed information only about a single torrent or debris flow system. Therefore, it is difficult to determine the spatial characteristics of a single large-scale debris flow event that triggers multiple debris

flows in the surrounding area. In addition, permanently installed monitoring systems often do not measure the deposition volumes, but use empirical relationships to calculate debris flow magnitudes as, e.g., in Comiti et al. [5]. However, accurate volume data are crucial for the calibration and validation of models [9,10]. Therefore, in recent years, debris flow transported volumes have increasingly been measured in more direct ways using highly-precision methods such as terrestrial laser scanning (TLS) [11,12] and airborne laser scanning (ALS) [13,14], or photogrammetric techniques using uncrewed aerial systems (UAS) [15,16]. In order to analyse the spatial effects and the accumulation volumes of a single debris flow event, pre- and post-event data acquisition is necessary (e.g., [17,18]). However, these data should not be collected at long intervals from the event, but must be carried out closely before and immediately afterwards. Otherwise, anthropogenic influences, as shown by Bull et al. [17], and erosion processes will confound the results. Due to these difficulties, there is a lack of studies analysing a single debris flow event in an alpine region with precise pre- and post-event data on catchment scale. In this study, we analyse the debris flows triggered by a heavy precipitation event in July 2022 in the Horlachtal catchment in the Stubai Alps. Despite the remote location and the high altitude of the valley, infrastructure such as hiking trails and hut supply roads were damaged and had to be rebuilt. For the study area, several high-resolution data sets are available for this event and for this catchment. Area-wide airborne LiDAR (light detection and ranging) topographic information based on our own ALS data acquisition 10 months prior to the event is available and can be compared with another LiDAR data set acquired only days after the debris flow initiations. These data allow us not only to investigate the event on the basis of individual debris flow channels, as in conventional monitoring systems, but also to conduct a unique spatial evaluation of the triggered debris flows, including their magnitude at catchment scale. Additional precipitation data from three different meteorological stations within the study area, as well as area-wide information from the gridded INCA (Integrated Nowcasting through Comprehensive Analysis) rainfall data [19], are also available. Together with discharge measurements from four different hydrological gauges, the spatial characteristics of the heavy rainfall events that led to the initiation of multiple debris flows can be further elaborated.

The results of this study are very useful for the calibration and validation of debris flow models [20], and thus for the establishment of spatial predictive models to identify potential future debris flow initiation sites. Such models include susceptibility models, as, e.g., those based on maximum entropy [21,22] or other machine learning methods [23], as well as logistic regression [24,25], to name a few.

## 2. Study Area

The Horlachtal is located within the Stubai Alps in Tyrol, Austria (Figure 1). It is a side valley of the Ötztal and extends from the village of Niederthai (about 1550 m) to elevations above 3300 m. Geologically, the Horlachtal belongs to the polymetamorphic Ötztal Crystalline of the Ötztal–Stubai Complex, with predominant paragneisses and orthogneisses, as well as mica schists [26,27], which strike in an east–west direction and thus are roughly parallel to the main valley [28]. The mean annual temperature at the meteorological station Horlachalm (1910 m) (Figure 1A) between 1991 and 2020 was 3.1 °C, while the mean annual precipitation total during this period was 820 mm [29]. The Horlachtal can be divided into six different sub-catchments. Besides the east–west striking main valley (HT), three parallel north–south oriented tributary valleys (Grastal GT, Larstigtal LT and Zwieselbachtal ZT), as well as the Weites Kar (WK) and the Finstertal (FT) in the northern part of the study area, can be distinguished. The main characteristics of each of these side valleys are listed in Table 1.

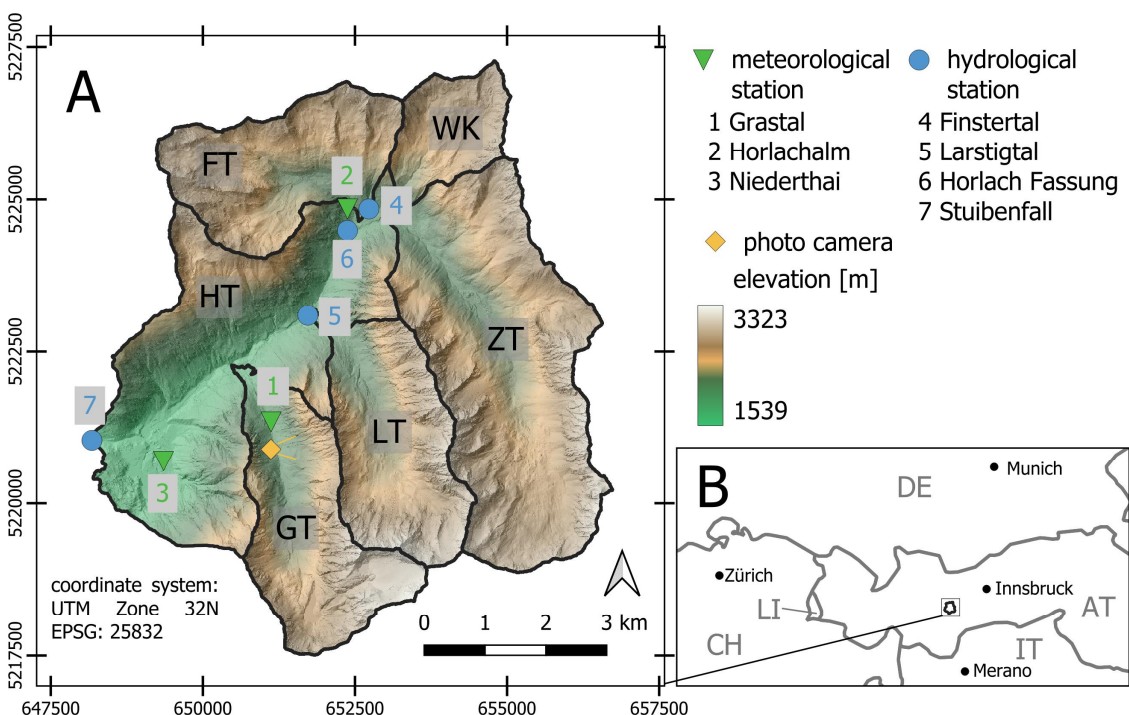

**Figure 1.** (**A**): The Horlachtal with its sub-catchments presented with a hillshade generated based on the 2022 LiDAR data. The locations of the stations mentioned are shown as well. (**B**): Location of the study area.

**Table 1.** The sub-catchments of the Horlachtal and their main properties.

| Sub-Catchment | Area [km$^2$] | Elevation Range [m] | Mean Slope [degree] | Number of Debris Flows between 1947 and 2020 * |
|:---:|:---:|:---:|:---:|:---:|
| HT | 15.407 | 1557–3001 | 33.8 | 88 |
| GT | 7.386 | 1702–3340 | 35.6 | 172 |
| LT | 7.046 | 1826–3340 | 36.5 | 197 |
| ZT | 15.06 | 2042–3241 | 33.1 | 304 |
| WK | 3.042 | 2050–3087 | 29.6 | 26 |
| FT | 6.95 | 1967–3061 | 31.5 | 47 |
| Total | 54.891 | 1557–3340 | 33.3 | 834 |

* Number of debris flows according to Rom et al. [29].

The debris flows in the whole or parts of the Horlachtal have already been the subject of previous studies [24,28–33]. In the vast majority, the debris flows occur as transport-limited slope-type debris flows, as classified and further explained in Wichmann [34] or Rickenmann and Zimmermann [35], with a hydrological catchment in the bedrock sections. During local heavy precipitation events in the summer [36,37], rainwater accumulates in these catchments and reaches the contact zone between the bedrock section and the adjacent talus slopes with high velocities [24]. If a certain rainfall intensity is exceeded, debris flows are initiated by the so-called "fire-hose effect" (see, e.g., [38]). Based on observations of large-scale debris flow events in the 1990s, this initiation threshold was set to about 20 mm per 30 min by Becht [28] and Becht and Rieger [32]. This comparatively low triggering intensity was attributed to the relatively low annual precipitation totals in the study area [31].

The geomorphologic features of an alpine catchment influence the spatial differences in debris flow susceptibility [39,40]. The landforms in Horlachtal were largely influenced by far advancing glaciers in the Pleistocene [41]. Today, however, the glaciers in the study area only cover about 1.5% of the total area. Especially along the sub-catchments GT, LT

and ZT, large talus cones have formed, covered with rockfall debris, which are often cut by debris flow channels up to 10 m deep. These landforms testify to a high morphodynamic on the slopes, especially in GT, LT and ZT. As shown in Rom et al. [29], most debris flow activity of the past decades is concentrated in these sub-catchments, with the majority of debris flows occurring on the west-facing slopes. This difference may be explained by the now deglaciated cirques that have formed on the east-facing slopes, preventing the accumulation of high peak discharges during heavy precipitation events [28,29].

## 3. Materials and Methods

In order to analyse the debris flow event in the Horlachtal catchment in July 2022, we used different data sets and methods. Figure 2 shows a technical roadmap of how the data sets are connected and interlinked. In this section, we will explain the pre-processing and evaluation steps of each of these data.

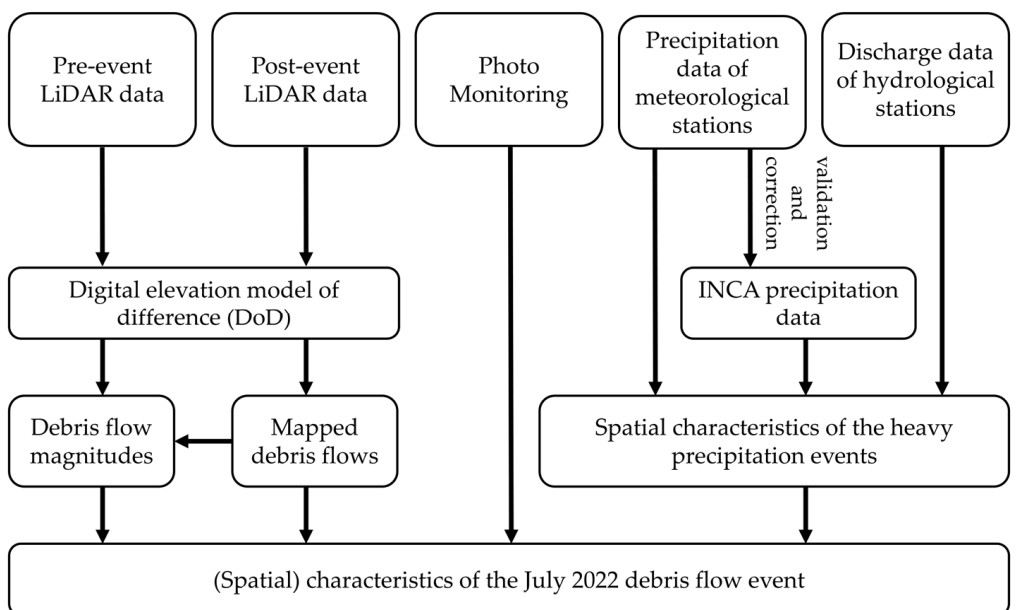

**Figure 2.** Technical roadmap of the study.

### 3.1. Photo Monitoring

To gain daily insights into the recent dynamics in the study area, we installed a Bolymedia MG983G-30M photo camera in GT in the summer of 2019 (see Figure 1A), overlooking one of the most active debris flow channels. This camera took a photo every day, which was sent to an online server in real time. Unlike in other debris flow monitoring systems with permanently installed field instruments [3,5,37,42], our camera could not switch to a high-frequency "event mode" [2], but was installed as a low-cost possibility to get daily impressions of the study area.

The July 2022 images showed newly accumulated debris flow material on two separate occasions (Figure 3). According to the images, a large debris flow event occurred between the photo acquisitions of 20 and 21 July (new debris flow deposition area outlined in blue), as well as a second, smaller debris flow event occurring between 22 and 23 July (debris flow deposition area outlined in red).

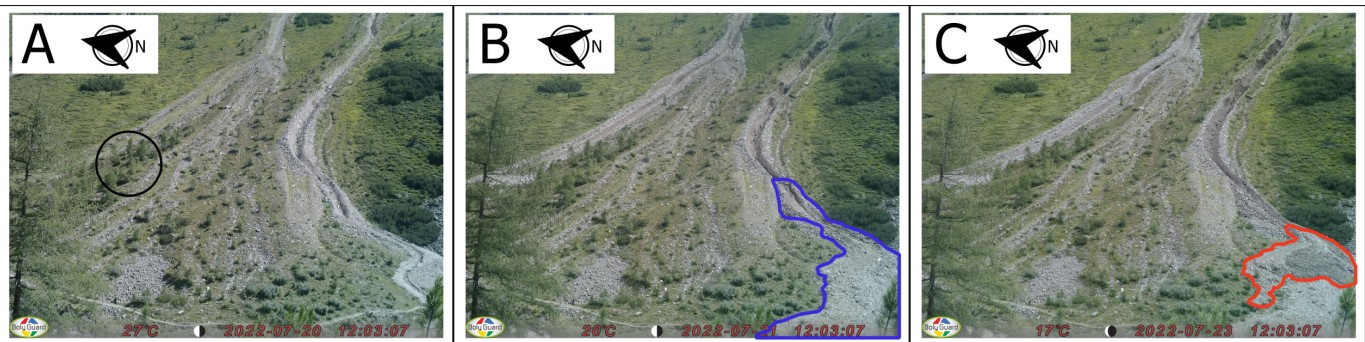

**Figure 3.** Photos taken by the camera in GT. (**A**): Pre-event photo from 20 July 2022. For scale: the encircled trees reach up to 10 m in height. (**B**): Photo after first event, dated 21 July 2022. (**C**): Photo after second event, taken on 23 July 2022.

*3.2. Topographical Data*

3.2.1. Data Acquisition

Two airborne LiDAR datasets were used to analyse the topographic changes caused by the debris flow event. The pre-event LiDAR data were acquired for the entire study area on 22 September 2021 by the Chair of Physical Geography of the Catholic University of Eichstätt-Ingolstadt using a Riegl VUX 1LR sensor integrated in a Riegl VP-1 HeliCopter-Pod (www.riegl.com; accessed on 13 February 2023), which was installed on an Airbus Helicopters H125 Ecureuil (Figure 4A). Due to the small and agile helicopter type, which is perfectly adapted to mountainous terrain, we were always able to maintain a constant height of about 100–150 m above ground. In addition to the airborne data acquisition, two different dGNSS ground stations were set up within the study area to record raw dGNSS data with a temporal resolution of 1 Hz (Figure 4B). These data were used in the post-processing steps in combination with an Inertial Measurement Unit (IMU) inside the VP-1 to reconstruct the flight trajectory accurately. The adjustment of the individual flight strips was done following Glira et al. [43] and Glira et al. [44] using the pointcloud processing software OPALS [45]. Subsequently, the GIS-software SAGA [46] with its extension LIS Pro 3D (Laserdata; [47]) was used to remove individual erroneous points and to classify the ground points, as described in Rom et al. [48]. The last step allowed us to generate a pre-event Digital Terrain Model (DTM) of the study area with a spatial resolution of $1 \times 1$ m.

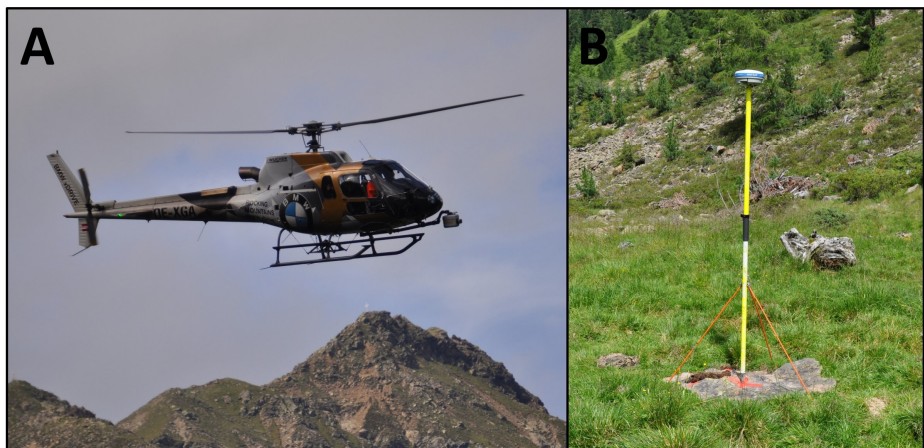

**Figure 4.** LiDAR data acquisition in Horlachtal. (**A**): VP1-HeliCopterPod mounted on an Airbus Helicopters H125 Ecureuil. (**B**): One of two dGNSS base stations on the ground.

The post-event LiDAR data acquisition was carried out on 3 August 2022 as a "special event-based data acquisition" and thus only two weeks after the main debris flow event

on 20 July 2022 in Horlachtal. Again, the entire study area was covered using the same acquisition setup as for the pre-event data (Table 2). All post-processing steps were carried out in the same way as described above. Finally, we were able to produce a high-resolution LiDAR-generated DTM before and after the debris flow event, covering all parts of the study area. Due to regular field surveys in between the two LiDAR data acquisitions and a monitoring system with a photo camera (see Section 3.1), we know that there was no major debris flow event in the Horlachtal between September 2021 and 20 July 2022.

**Table 2.** Setup parameters for the pre-event (2021) and post-event (2022) LiDAR data acquisitions.

| Acquisition Date | Type | Platform | Scanner Model | Mean Point Density [pts/m$^2$] | Pulse Repetition Rate [kHz] |
|---|---|---|---|---|---|
| 22 September 2021 | ALS | Airbus Helicopters H125 Ecureuil | Riegl VUX 1LR | 16.1 | 200 |
| 3 August 2022 | ALS | Airbus Helicopters H125 Ecureuil | Riegl VUX 1LR | 24.4 | 200 |

3.2.2. DTM of Difference and Debris Flow Volumes

By subtracting the DTM 2021 from that of 2022, we calculated the DTM of Difference (DoD), which shows the topographic changes between the two data acquisitions. As typical debris flow process landforms such as transport channels, levées or accumulation zones become visible in the DoD, we were able to map the debris flow process areas of the July 2022 event. In order to improve the spatial quality of the DoD, we performed co-registration steps before calculating the volumes of the debris flow accumulations. Due to the large amount of data, and therefore to reduce processing time, we defined 23 smaller areas surrounding the debris flow process areas (see Figure A1 in Appendix A). For each of these subsets, the two LiDAR-derived datasets could be registered at a local scale. First, we mapped stable areas with similar topographic characteristics (slope, aspect) as the process areas, in which no geomorphological changes between the two data acquisitions were to be expected. Here, Iterative Closest Point (ICP) algorithms integrated in LIS Pro 3D [49] were used for pointcloud-based co-registration. In addition, the raster-based approach of Nuth and Kääb [50], implemented in the Python package pybob, was used as a second co-registration method.

After co-registration, all debris flow process areas could be mapped in great detail based on the DoD. For each accumulation area, the DoD cell values were summed to calculate the deposited volume of the corresponding debris flows.

3.2.3. Error Assessment of the Debris Flow Volumes

To get a first idea of the errors contained in the DoD, the values in the mapped stable areas of each of the 23 different subset areas could be evaluated. Table A1 (in Appendix A) provides statistical measures of the quality of the calculated model, including precision (standard deviation) and accuracy (root mean square error – RMSE). For each DoD-derived accumulation volume, the error assessment was calculated according to Anderson [51]. This approach combines the uncorrelated random error, the spatially correlated random error as well as the systematic error.

*3.3. Meteorological Data*

Precipitation data were available from three different meteorological stations within the study area (Table 3; every coordinate information throughout this paper refers to UTM zone 32N – EPSG 25832). All of them recorded without issues during the intense rainfall events in July 2022. The Grastal station is operated and maintained by the SEHAG research group. The data from the meteorological stations Horlachalm and Niederthai were provided by the Tyrolean Hydropower Company (TIWAG). As the three stations are distributed within the study area (Figure 1A), we can evaluate possible spatial differences in rainfall intensities during the heavy precipitation events.

**Table 3.** Site parameters of the meteorological stations used within this study.

| Station Name | Operator | Easting [m] | Northing [m] | Altitude [m] | Temporal Resolution [min] |
|---|---|---|---|---|---|
| Grastal | SEHAG | 651,120 | 5,221,345 | 2000 | 10 |
| Horlachalm | TIWAG | 652,550 | 5,224,656 | 1968 | 15 |
| Niederthai | TIWAG | 649,354 | 5,220,696 | 1615 | 15 |

To understand the spatial differences in precipitation during the debris flow event better, we used INCA data provided by the Austrian Central Institute for Meteorology and Geodynamics (ZAMG). This multivariable and nowcasting system uses measurements from meteorological stations as well as radar and satellite data in combination with topographic information to provide area-wide datasets for multiple parameters (including precipitation), and has been specifically developed for mountainous terrain [19]. The gridded INCA data are available with a temporal resolution of 1 h and a spatial resolution of $1 \times 1$ km.

As INCA precipitation data are likely to overestimate peak rainfall values for extreme events [52], we used the recorded data from the meteorological stations in the study area to validate and correct the absolute values of the gridded INCA data. This was done by establishing a relationship through linear regression between the INCA cell values and the recorded precipitation values of the stations.

### 3.4. Hydrological Data

In addition to the meteorological stations, four different hydrological gauges provided discharge information for the July 2022 events (Figure 1A). The Finstertal and Larstigtal gauges are maintained by the SEHAG research group, while the gauges Horlach Fassung and Niederthai are operated by the TIWAG (Table 4). In contrast to the meteorological stations, the measurements of the hydrological gauges do not only refer to the exact location of the instrument, but also represent the entire hydrological catchment area of the installation. Thus, depending on their location, the gauges record the discharge of one or multiple sub-catchments (Table 4). However, most of the water at the Horlach Fassung gauge is diverted to the Finstertal reservoir in a neighbouring valley. These diversions in turn affect the measurements at the Stuibenfall gauge further down the valley.

**Table 4.** Site parameters of the hydrological gauges used within this study.

| Station Name | Operator | Easting [m] | Northing [m] | Altitude [m] | Temporal Resolution [min] | Sub-Catchments Captured |
|---|---|---|---|---|---|---|
| Finstertal | SEHAG | 652,558 | 5,224,720 | 1975 | 15 | FT |
| Larstigtal | SEHAG | 651,725 | 5,223,097 | 1828 | 15 | LT |
| Horlach Fassung | TIWAG | 652,550 | 5,224,656 | 1968 | 15 | FT, WK, ZT |
| Stuibenfall | TIWAG | 648,176 | 5,221,042 | 1533 | 15 | all * |

\* Influenced by the amount of water diverted at Horlach Fassung.

The hydrological stations operated by SEHAG (Finstertal and Larstigtal) consist of Ott CTP probes that record water levels with a resolution of 15 min. Water level is converted to discharge using a site-specific rating curve (i.e., second-degree polynomial function), which describes the non-linear stage–discharge relationship. Regular discharge measurements were carried out using the salt dilution method during the period 2019–2022. At Finstertal gauge, we used an automated salt injection system developed by Fathom Scientific (https://www.fathomscientific.com/; accessed on 13 February 2023) to collect event-based discharge measurements. The collection of infrequent discharge events allowed us to reduce the uncertainty in the extrapolation range of the rating curves. The highest discharge measured was 3.04 m$^3$/s at Larstigtal (water level 0.46 m) and 1.94 m$^3$/s at Finstertal (water level 0.41 m). The stream gauge at Larstigtal was destroyed by high debris load of the river

during the debris flow event on 20 July 2022 and had to be rebuilt after the event at another site further upstream.

## 4. Results

### 4.1. Meteorological Analyses

Precipitation data were recorded at three different meteorological stations in the study area (Horlachalm, Grastal, Niederthai). In Figure 5, the recordings are shown aggregated to 30 min sums. All stations recorded heavy rainfall at the time of the main debris flow event in the afternoon of 20 July 2022. Here the data show a short precipitation event lasting about 2 h, from 4 p.m. to 6 p.m. Expressed as a convective thunderstorm, this event appeared very sudden, as no antecedent rainfall could be registered in the hours (and even days) before and after. The last minor precipitation prior to the event was recorded on 7 July and therefore 13 days earlier. Another aspect that becomes apparent in Figure 5 is that the extreme rainfall event of 20 July shows spatial differences in the amount and intensity of precipitation data between the stations. The highest peak of the 30 min sum was recorded at the Horlachalm station (17.8 mm), but the total sum of the event was about the same compared with the Grastal station (27 mm). The Niederthai station, however, recorded significantly less precipitation for this event. With a total of 12.4 mm, it had only half the precipitation compared with the other two stations.

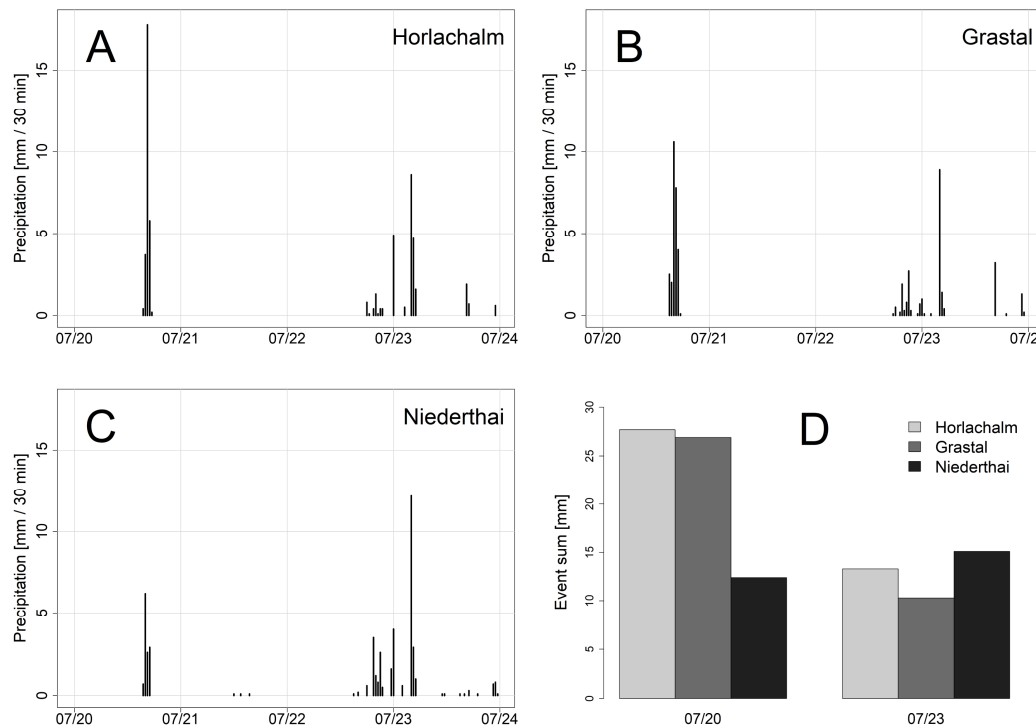

**Figure 5.** Precipitation intensities recorded by the meteorological stations Horlachalm (**A**), Grastal (**B**) and Niederthai (**C**). The barplots on the bottom right (**D**) display the total precipitation sums of the two events on 20 July (4 p.m. to 6 p.m.) and on 23 July (4 a.m. to 5 a.m.).

During the evening and night of 22 July, some smaller amounts of rainfall were recorded at the stations, but the second precipitation event that triggered some small debris flows occurred on Saturday, 23 July 2022. This was again a short but heavy rainfall event lasting from about 4 a.m. to 5 a.m. The Horlachalm and Grastal stations showed similar patterns, with a peak of about 8.5 mm/30 min and a total event sum of 10 to 13 mm. This time, the Niederthai station recorded the highest peak for 30 min (12.2 mm) and the total sum was the highest of all stations (15.1 mm).

Overall, a larger amount of rainfall could be detected for the event on 20 July over the whole study area. However, this picture was not uniform for the whole Horlachtal, as is shown by the Niederthai station.

The results from the meteorological stations already suggest some spatial differences of the precipitation events. In addition, the INCA data can provide an even higher spatial resolution picture of the extreme events. Figure 6 shows the INCA data of the main precipitation event on 20 July, together with the corresponding meteorological station measurements. These data show that the convective cell responsible for initiating the debris flows suddenly appeared at a local scale at 4 p.m., before disappearing again. Although this behaviour is also shown by the station measurements, the amount of precipitation differs between the two datasets.

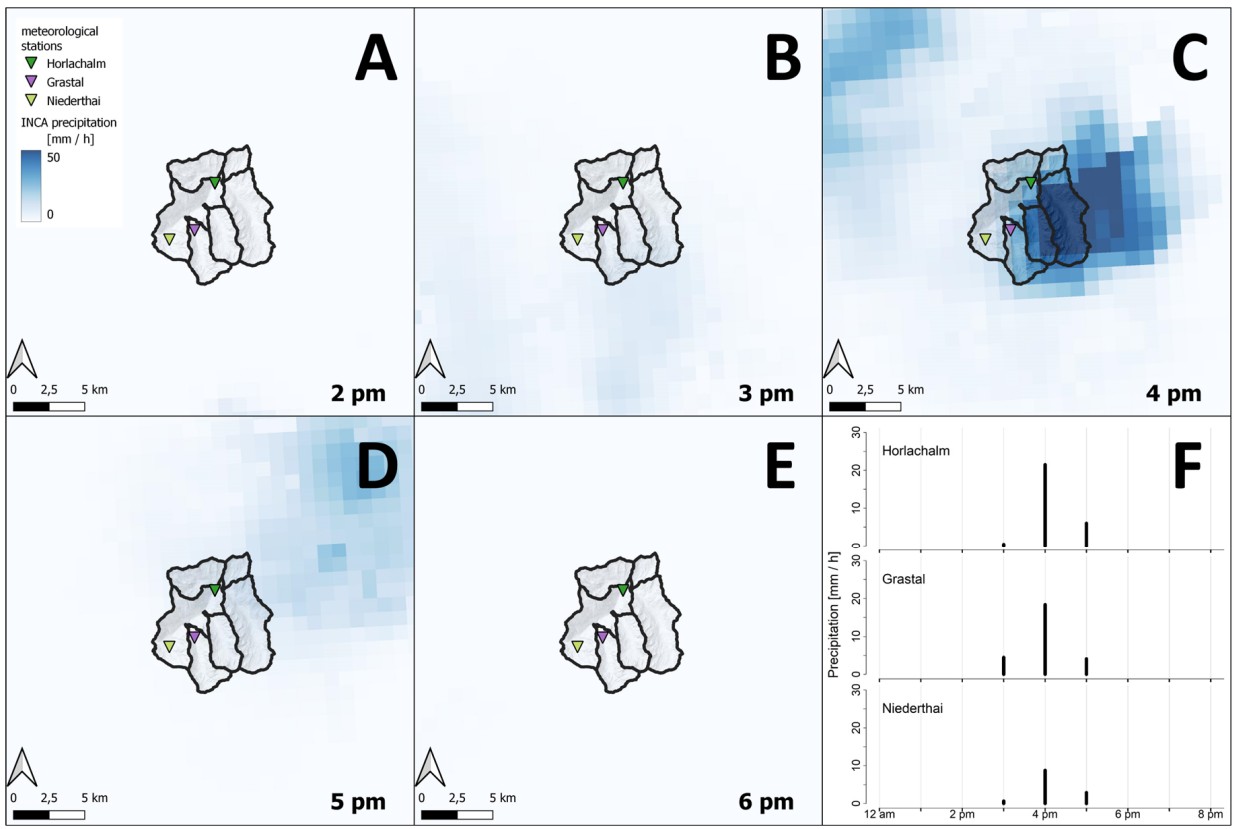

**Figure 6.** (**A–E**): The precipitation event on 20 July 2022 as detected by the INCA system. For comparison, the graphs in (**F**) represent the measurements of the meteorological stations in Horlachtal.

Although the number of meteorological stations in the study area is very limited for a linear regression and therefore prone to uncertainties, we detected a linear relationship between the INCA values and the rainfall measurements at the three stations, with the aforementioned and expected slight overestimation of the amount of rainfall by the INCA data [52]. Figure 7A shows the recorded precipitation of the three stations during the rainfall event on 20 July 2022 between 4 p.m. and 5 p.m., compared with the corresponding INCA cell values of the same timeframe. The calculated linear model showed a strong correlation between the two data sources ($R^2 = 0.97$) and was therefore used to correct the INCA cell values during this precipitation event (Figure 7B).

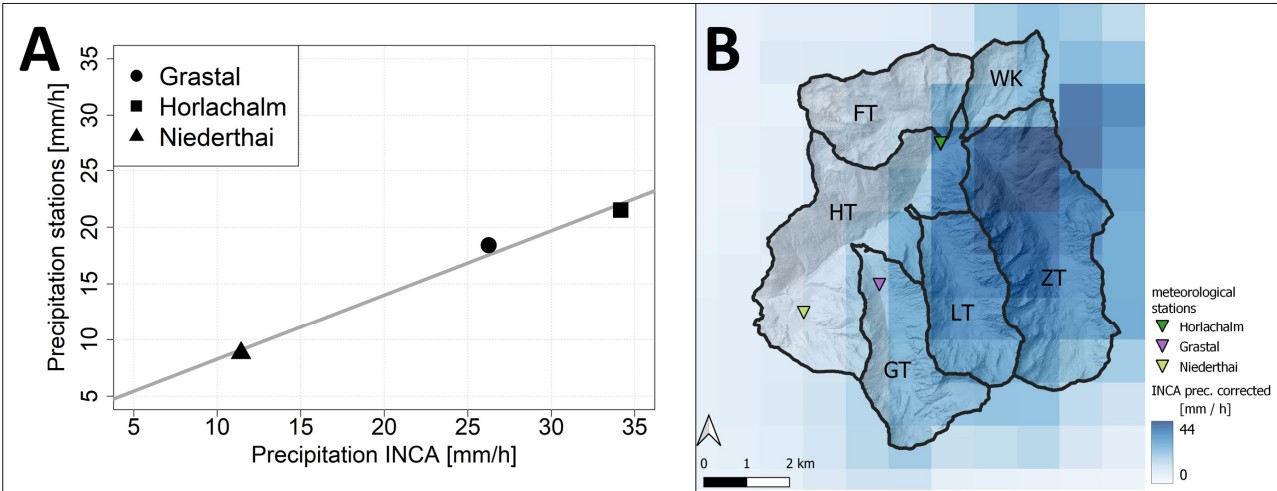

**Figure 7.** (**A**): Relationship between the precipitation measurements at the meteorological stations and the INCA rainfall data for the event on 20 July 2022, from 4 p.m. to 5 p.m. (**B**): Gridded INCA data with corrected absolute precipitation values for the same timeframe.

The area-wide precipitation map at the peak of the precipitation event between 4 p.m. and 5 p.m. now shows that the extreme rainfall event had the highest values in the lower ZT (up to 44 mm). Besides ZT, the sub-catchments LT and GT were also largely affected, with precipitation values ranging from 20 mm (west-GT) to 40 mm (east-LT). The intensities in WK showed a high variability, with medium-to-high values (13 mm to 24 mm), while FT and HT were rather less affected. In these valleys, the precipitation values mostly ranged between 8 mm and 20 mm.

### 4.2. Discharge Measurements

Discharge was measured at four different locations in the study area during the July 2022 rainfall events (Figure 8). However, we cannot be sure that the maximum discharge was measured accurately because of the heavy bedload and the fact that the streams overflowed their beds in some places. In addition, the rating curves were not calibrated for such high water levels. Nevertheless, the two events of 20 and 23 July were recorded at each gauge, with a sudden increase in the discharge values within a very short time (~1 h) and a slower, but still rapid, decrease after the rainfall events. At the Finstertal gauge (Figure 8A); however, the peak discharge values for both events (0.5 m$^3$/s with 0.26 m water level and 0.64 m$^3$/s with 0.29 m water level) were not exceptionally high compared with the records of the previous three years. Since the establishment of this gauge in June 2019, these values have been exceeded on several occasions, e.g., on 20 June 2021 (2.5 m$^3$/s with 0.48 m water level) or on 27 June 2020 (1.12 m$^3$/s with 0.35 m water level). Discharge at Horlach Fassung (Figure 8B) showed larger amplitudes, especially for the event on 20 July, with values close to 4 m$^3$/s. The Larstigtal gauge (Figure 8C) also recorded a very sudden and strong increase during the first event, with an estimated peak discharge of 3.95 m$^3$/s (0.5 m water level). However, this strong increase was apparently associated with strong turbulence, as the gauge broke at this point, probably due to coarse bedload transport. Thus, all detected discharge values recorded at the Larstigtal gauge after the peak on 20 July (see Figure 8) are erroneous. The hydrological station in Niederthai (Figure 8D) drains the entire study area. Here, the discharge reached 8.3 m$^3$/s on 20 July, which was almost twice as high as the peak of the second event (4.2 m$^3$/s). The unsteady flow behaviour at the Niederthai gauge before and after the events was caused by the management of the diversion at Horlach Fassung.

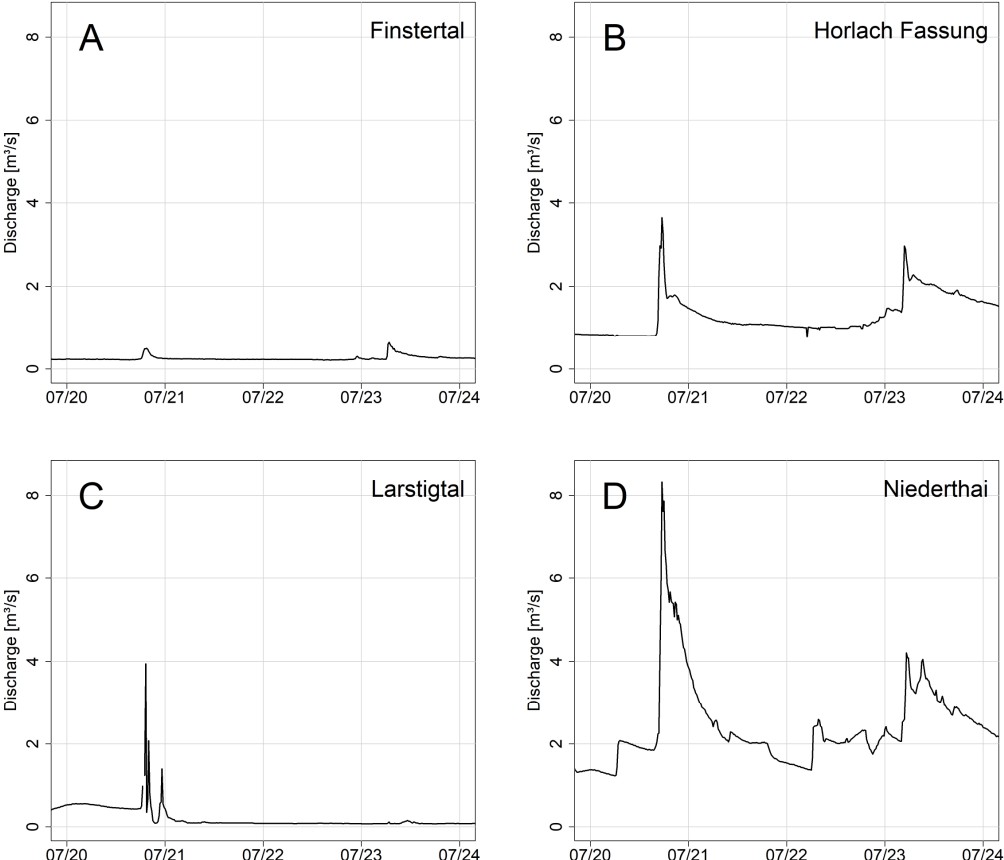

**Figure 8.** Discharge measurements at the four gauges (**A**–**D**) in Horlachtal during the events in July 2022.

### 4.3. Evaluation of Topographical Data

In the calculated DoD, the topographic changes due to the debris flows of July 2022 become visible. Figure 9 shows an example of a debris flow in LT. The starting point of the debris flow is at the contact zone between the talus cone and the steep bedrock section above (hydrological catchment). The red DoD cells in the transit zone indicate erosion and incision into the talus slope. Right before the transition into the accumulation area (blue), levées have formed on both sides of the transit channel. Typical landforms of past debris flows are also visible in the underlying hillshade.

#### 4.3.1. Spatial Patterns

In total, we were able to map 156 different debris flows associated with the July 2022 events with the help of the DoD. However, the spatial variability is very high (Figure 10). Most processes were initiated in GT (especially on the west-facing slopes), in LT and on the east-facing slopes of ZT. Only a few debris flows occurred in HT, on the east-exposed slopes in GT and on the west-facing slopes in ZT. No debris flows were triggered in FT and WK.

Figure 11 shows the total number of triggered debris flows in the respective sub-catchments. However, as the different sub-catchments vary in size, we have also normalised the number of debris flows for each square kilometre. Although the total number of detected processes was highest in ZT, more debris flows were initiated in LT when we related them to the area.

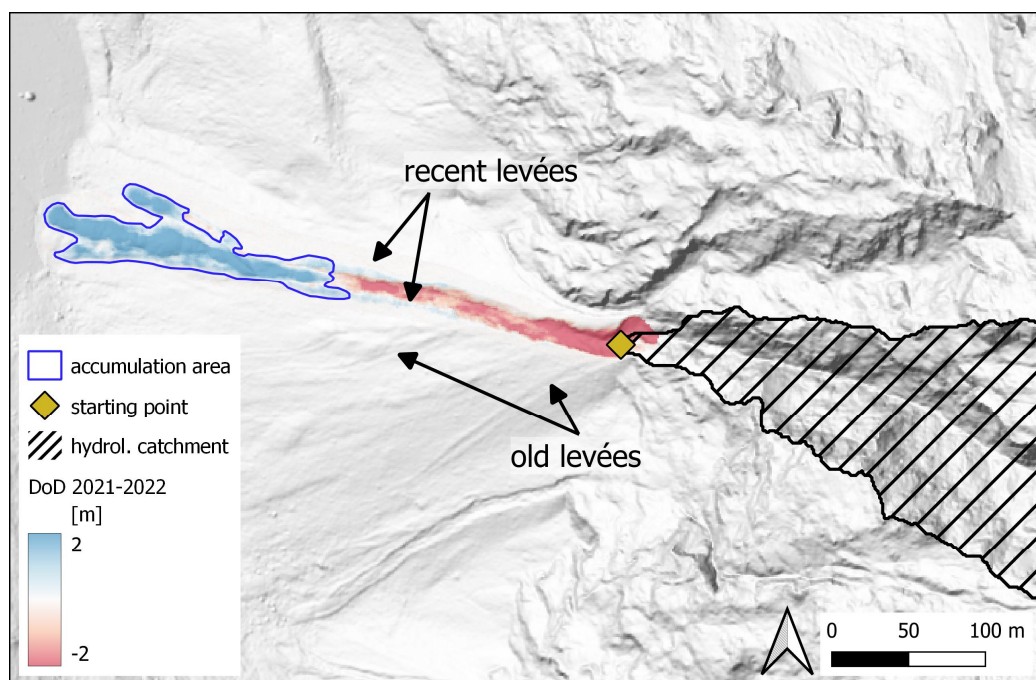

**Figure 9.** Example of a detected debris flow in LT (E: 653275, N: 5220945 (EPSG: 25832)) based on the DoD.

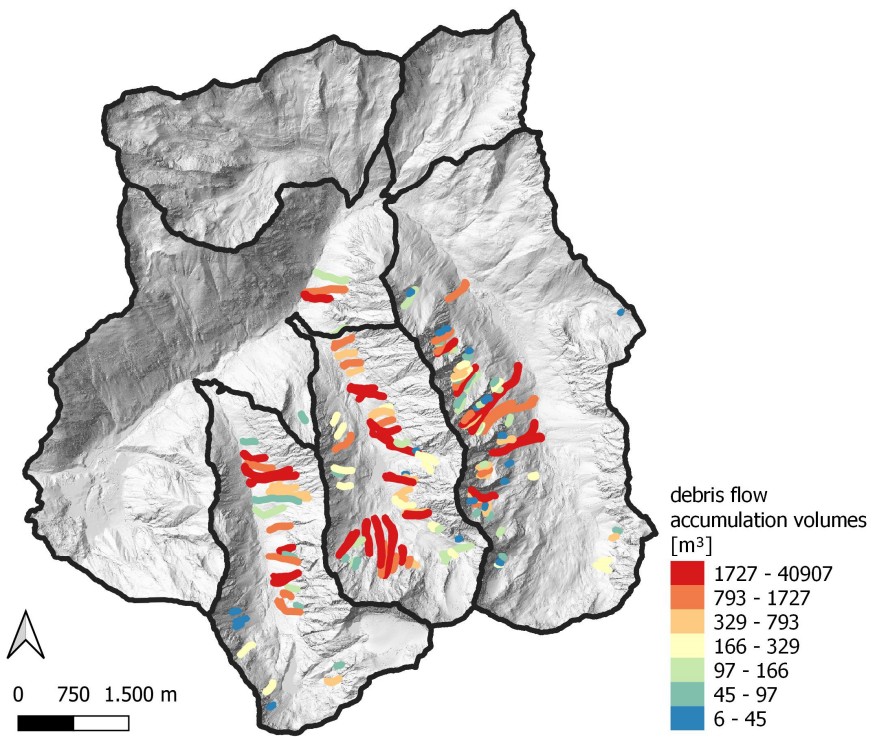

**Figure 10.** Spatial distribution of the 156 detected debris flows triggered in July 2022, classified by their accumulation volumes.

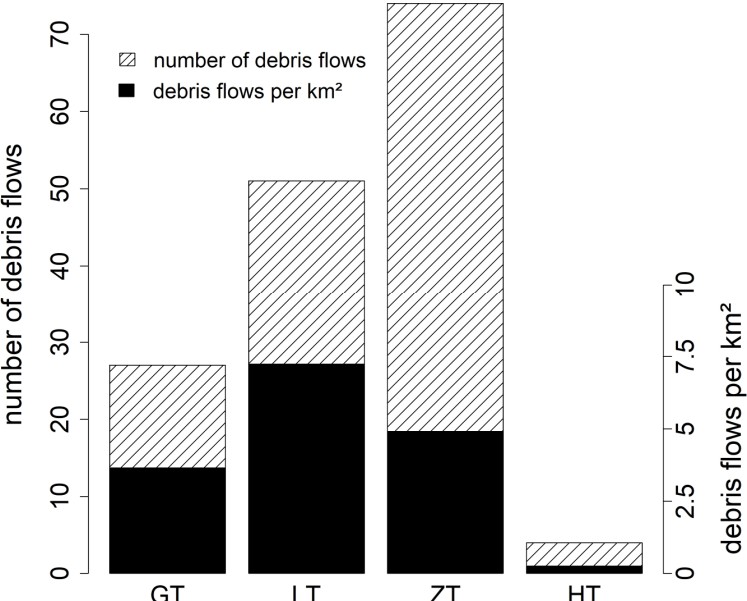

**Figure 11.** Number of debris flows per sub-catchment as well as the normalised debris flow count per square kilometre for the respective sub-catchments.

### 4.3.2. Debris Flow Accumulation Volumes

The triggered debris flows were of very different magnitudes, with accumulation volumes ranging from 6.4 m$^3$ to almost 41,000 m$^3$ (see Figure 10). A total of 199,267 m$^3$ of debris flow material was deposited during the two events in July 2022 in Horlachtal. Figure 12 displays how the volumes were distributed among the sub-catchments. In each of the areas, debris flows of lower and higher magnitude occurred. However, there were many large volumes recorded in LT, including the largest one (40,907 m$^3$), which alone deposited more debris than, for example, all GT debris flows together (37,869 m$^3$). Another interesting point was the relatively low total volume in ZT compared with the very high number of debris flows initiated there. Thus, we can register many small accumulations for ZT.

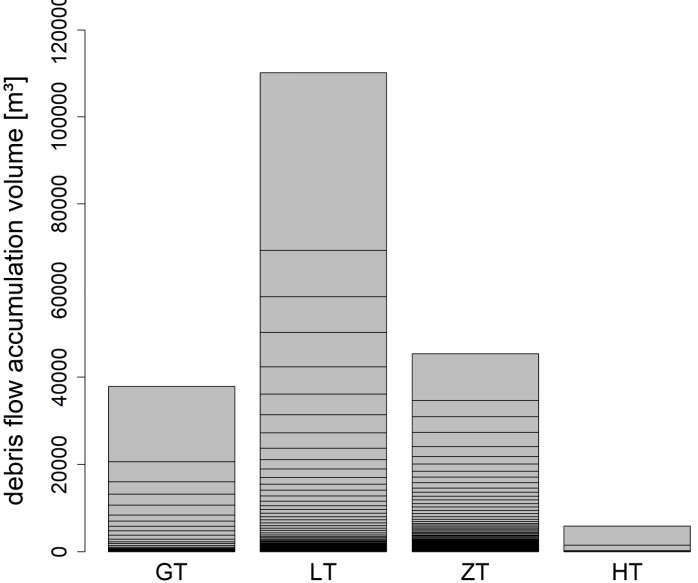

**Figure 12.** Debris flow accumulation volumes separated between the sub-catchments. Each single debris flow is represented by a rectangle with a size according to the accumulation volume.

## 5. Discussion

### 5.1. The Dating of the Debris Flow Event

The images of the automatic camera show that the debris flows of July 2022 were triggered on two separate occasions. In combination with meteorological and hydrological data, the photos date the initiations to Wednesday, 20 and Saturday, 23 July. The recorded data from the hydrological gauge at Niederthai and the precipitation totals from the meteorological stations indicate a greater impact of the first event. This is consistent with the images of the photo camera, which clearly show the larger debris flow event on Wednesday. Therefore, from the available data, the main debris flow event is dated to the afternoon of Wednesday, 20 July 2022. The second, smaller event recorded by the camera on Saturday is less easy to interpret, as we only know of it from the one debris flow channel shown on the camera. Using the area-wide LiDAR data, we cannot distinguish between two debris flow lobes of the same precipitation event, or two successive triggering events for all other debris flow channels.

### 5.2. Spatial Differences of the Debris Flow Events

The most important factor influencing the spatial distribution of the debris flows that occurred in July 2022 in the Horlachtal was the spatial pattern of the extreme precipitation event on the 20 July, which served as the main debris flow triggering event. In addition to sufficient precipitation intensity, the presence of a hydrological catchment was also crucial for the initiation of slope-type debris flows in Horlachtal [28]. Figure 13A shows the Horlachtal with its side valleys. All debris flow catchments that were active at least once between 1947 (see [29]) and 2022 are integrated. Based on the analysed airborne LiDAR data, we can distinguish between catchments that initiated a debris flow in July 2022 and catchments that were not triggered during the 2022 event.

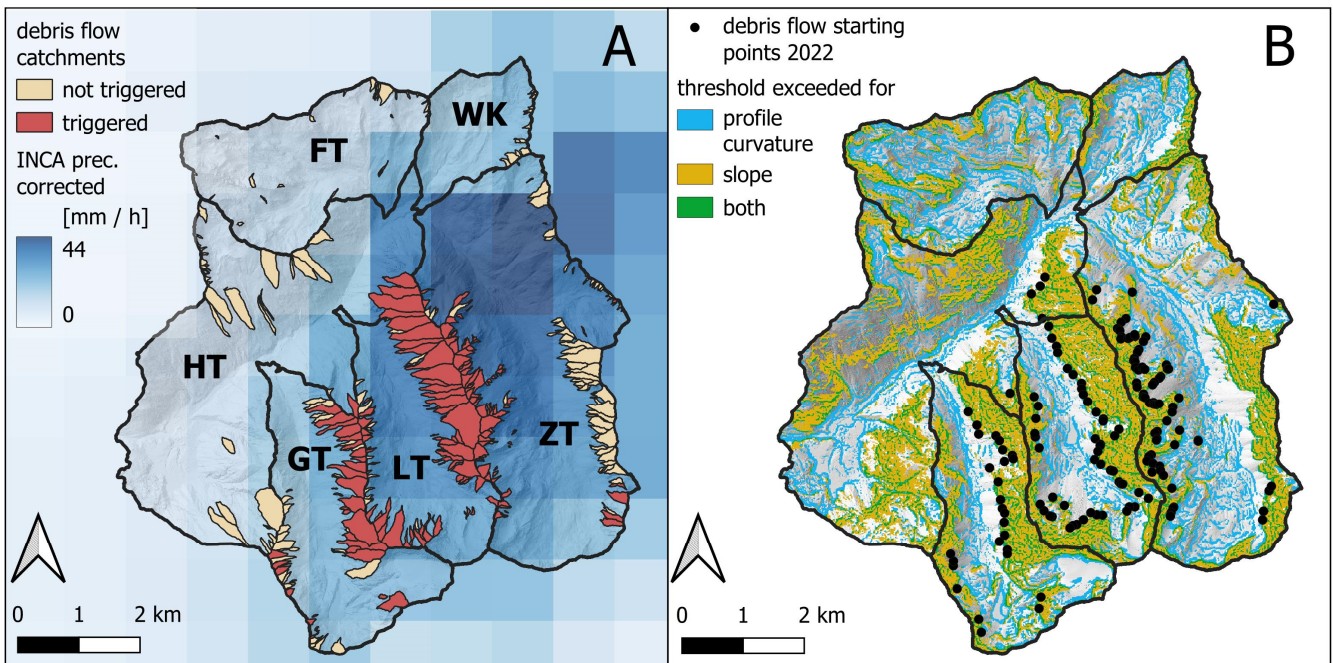

**Figure 13.** (**A**): The mapped debris flow catchments in Horlachtal, which have been active at least once since 1947. Catchments that did not trigger a debris flow in July 2022 (brown) can be distinguished from catchments that did (red). For discussing the spatial distribution of the initiated debris flows, the INCA rainfall data (with corrected absolute values) of the event on 20 July is depicted as well. (**B**): Areas for which the relief parameters slope and profile curvature exceeded a threshold value. These thresholds were set to the values of the 75th percentile of the starting points of the 156 debris flows in July 2022 for the respective parameter.

The spatial pattern of the INCA rainfall data can explain the absence of debris flows in FT, WK and large parts of HT. The results of the hydrological records further support the hypothesis that the spatial distribution of the precipitation intensities was the crucial factor for the spatial differences in triggered debris flows. This is supported by the comparatively small peak in discharge at the Finstertal gauge. Thus, all available datasets indicate a very local thunderstorm as the debris flow triggering event, as explained for example in Underwood et al. [36] or Berti et al. [37], which mainly affected the debris flow systems in GT, LT and ZT.

Figure 13 additionally shows that aspect was not a decisive factor for the spatial distribution of debris flows during the 2022 event. In GT, most processes were initiated on west-facing slopes. In contrast, almost all debris flow systems in LT were triggered, indicating a possible peak of rainfall intensity in LT. In ZT, debris flows were almost exclusively triggered on east-facing slopes.

However, other relief parameters apart from aspect play an important role for debris flow initiation as well [24,39,53,54]. According to Becht [28], the slope angle is of particular importance for triggering debris flows in Horlachtal. For the type of debris flows that occurred in the study area, Rickenmann and Zimmermann [35] determined a minimum slope of 27° at the starting zone. For the processes triggered in July 2022, the slope at the mapped starting points ranges between 28.4 and 55.4° (calculated based on the DTM of 2022 resampled to a cell size of $10 \times 10$ m) and is thus above the threshold in all cases. In addition, as explained in Heckmann et al. [24], a negative profile curvature (concavity in the long profile) increases the probability for the initiation of debris flows. In the case of the July 2022 event, a negative profile curvature (based on the $10 \times 10$ m DTM) could be detected at 154 out of 156 starting points (98.7%). Due to the strong influence of slope and profile curvature on debris flows initiation, we calculated thresholds for both parameters based on the observations of the debris flows in July 2022. These thresholds were set at the 75th percentile of the respective parameter using the values of the 156 debris flows detected with the help of the DoD. Figure 13B shows where the thresholds were exceeded in the study area, e.g., large parts of GT, LT, ZT and HT. However, to gain more information on the spatial distribution of potential debris flow starting points, detailed geomorphological mapping can be included in further studies, e.g., to localise sediment source areas.

Due to the area-wide availability of precipitation values based on the relationship between station measurements and INCA data, we can also include the spatial distribution of absolute precipitation intensities into our analyses. However, as hail was involved in the extreme event (according to a local farmer), the event sum recorded at the meteorological station might be underestimated. On the other hand, hail does not produce immediate runoff in the catchments and is therefore less involved in debris flow initiation.

Considering the mean rainfall intensity at each individual catchment for the crucial hour of the July event (Figure 14), we can compare them by triggered and non-triggered catchments. In all sub-catchments, with both triggered and not-triggered debris flow systems, the median rainfall intensity of the triggered catchments was higher than that of the non-triggered catchments. This difference is most pronounced for HT. In FT, rainfall intensities at every debris flow catchment are likely to be well below the proposed threshold of 20 mm/30 min [28,32].

Figures 13A and 14 show that the spatial pattern of the corrected INCA precipitation event can explain some of the differences in debris flow occurrence. However, some inaccuracies remain. The most striking are the west-facing slopes in ZT, where no debris flows were triggered despite high INCA rainfall intensities. A possible explanation could be that some debris flow systems temporally changed from transport-limited to material-limited systems, as explained in Rom et al. [29]. This would especially be the case at deeply incised and frequently triggered debris flow channels, and could occur as a result of channel recharge rates [55,56]. However, this is a rare phenomenon at individual locations and therefore cannot explain why all debris flow systems on the west-facing slopes in ZT were inactive during the 2022 event. A much more reasonable explanation is slight inaccuracies

in the INCA dataset. In general, the estimation of spatially high-resolution rainfall data can be very challenging [57]. Even the use of radar systems introduces uncertainties, e.g., due to beam blockage, especially in mountainous areas [58,59]. As shown by Ghaemi et al. [52], INCA rainfall datasets can also be inaccurate. In comparison with WegenerNet reference data, they prove that the peak of INCA precipitation intensities as well as the rainfall durations can show differences for extreme convective short-duration events. In addition, the spatial pattern may also differ from the reference data [52]. Therefore, it is possible that the spatial pattern of the INCA rainfall dataset is slightly shifted in the case of the extreme event on 20 July 2022. This is clearly illustrated by comparing the spatial patterns of the precipitation data and the active debris flows in Figure 13A. Another source of uncertainty in precipitation intensities is the hourly temporal resolution of the INCA data. For short-duration precipitation events, such as those discussed in this study, hourly time steps may be too coarse and introduce uncertainties into the rainfall intensities.

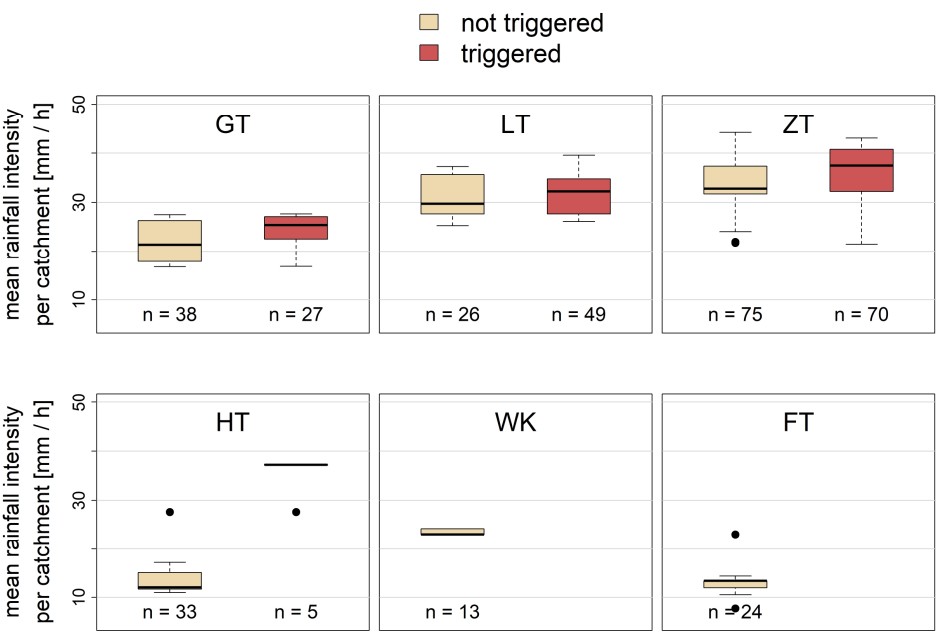

**Figure 14.** Comparison of the mean rainfall intensity for each debris flow catchment during the rainfall event on 20 July 2022 divided into the different sub-catchments.

The results of analyses on the spatial differences of individual debris flow events as presented here can be compared with predictive spatial models. Using multivariate logistic regression, Heckmann et al. [24] established a debris flow susceptibility model for LT, ZT and WK, which fits to the spatial characteristics of the July 2022 event for most parts. The main exceptions are the west-facing slopes of ZT and the eastern parts of WK. For these areas, the model in [24] predicts a high susceptibility, but no debris flows were triggered in July 2022. This indicates that predictive models (e.g., [21,22,60]) cannot fully explain the spatial effects of a single event.

### 5.3. Debris Flow Magnitudes

The debris flow catchments are not only an important factor for the spatial distribution of slope-type debris flows, but their morphometry is also known to influence the corresponding debris flow magnitudes [61–64]. This phenomenon has also been observed in the Horlachtal, where the size of the catchment is one of the most important parameters correlating with the debris flow volumes [29,32]. Since we have information on the spatial distribution of the precipitation event, we can use the total amount of rainfall for each of the debris flow catchments triggered during the 2022 event. Figure 15 shows the relationship between the amount of rainfall and the corresponding debris flow accumulation volumes

of the debris flows of July 2022 in Horlachtal. The calculated correlation coefficient Spearman's rho of 0.6 is slightly higher than the relationship between volume and catchment area in Rom et al. [29] (rho = 0.46) and in a similar range compared with De Haas and Densmore [65] (relationship between volume and catchment area: r = 0.63) and Franzi and Bianco [66] (rho = 0.639). A p-value lower than 0.001 indicates a highly significant relationship between the parameters. Using the 5% quantile regression line and the 95% quantile regression line (Figure 15) as lower and upper bounds, we can estimate the range of possible debris flow volumes for rainfall amounts. In the case of the 2022 Horlachtal event, the minimum accumulation volume is derived by

$$V_{min} = 0.295 \times r^{0.742} \tag{1}$$

for the required amount of rainfall r. The upper bound and therefore the maximum accumulation volume can be calculated by

$$V_{max} = 15.1 \times r^{0.854} \tag{2}$$

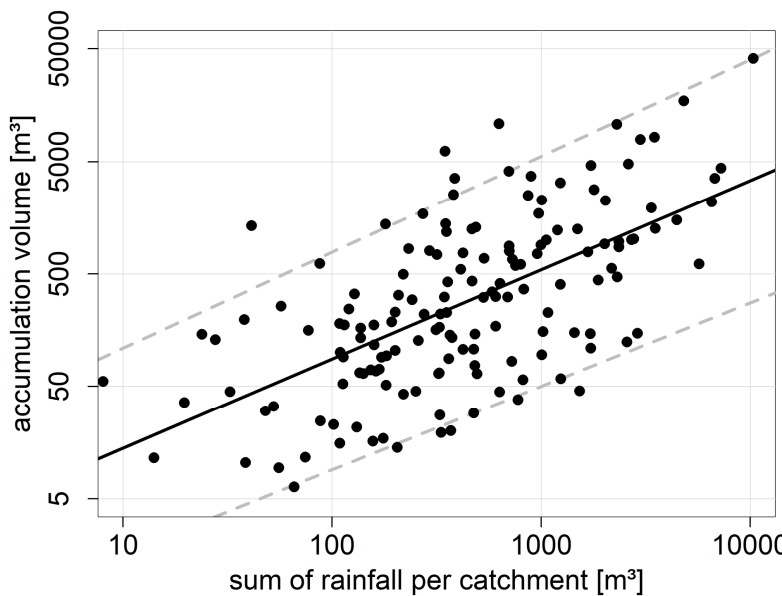

**Figure 15.** Relationship between the accumulation volume of triggered debris flows and the total sum of precipitation in the respective catchments during the event on 20 July 2022. The grey dashed lines represent the 5% and 95% quantile regressions.

Despite the rather large possible ranges of debris flow volumes (differences by a factor of 100 for the same amount of rainfall), this relationship may be useful for modelling future debris flow volumes in the Horlachtal. Furthermore, debris flow runout distances can be calculated from modelled volumes [67,68], which is, e.g., crucial for hazard modelling [69].

The remaining uncertainties in the simplified model between accumulation volume and precipitation totals for catchments may be due to the shift in the INCA data discussed above. In addition, there may be other parameters that play an important role in predicting debris flow volumes apart from the size of the catchment in combination with the precipitation event [66]. Other morphometric parameters of the catchment may play an accentuating role, as well as factors such as the availability of suitable debris flow material at the starting zone or hydrological properties.

## 6. Conclusions

In the summer of 2022, the Stubai Alps were affected by large-scale debris flow events. The installed photo camera in combination with hydrological and meteorological data showed that a local thunderstorm with high rainfall intensities (>20 mm/h) triggered

multiple debris flows in the Horlachtal on 20 July. A second, smaller event was recorded on 23 July.

Three different meteorological stations revealed great variability in the rainfall intensities recorded during the extreme event. This demonstrates the local character of such convective events, especially in alpine environments. From the gridded INCA rainfall data, we also know that it is likely that none of the three stations recorded the peak intensity of the event. It is therefore almost impossible to determine the rainfall intensities required to trigger debris flows based on precipitation data recorded at a certain distance from the process areas. However, using the recorded rainfall amounts to correct the INCA rainfall magnitudes, we can set the maximum intensity of the thunderstorm at 44 mm/h.

The DoD based on pre- and post-event LiDAR data reveals that the event initiated 156 different debris flow processes in the study area. However, there were large spatial differences throughout the Horlachtal, as most debris flows were triggered in the LT, as well as on the east-facing slopes in the ZT and the west-facing slopes in the GT. The INCA rainfall data as well as relief parameters can only partially explain this pattern.

Limitations include inaccuracies in the radar information, which indicate a slight spatial offset of the INCA data. A small spatial translation of the precipitation event (e.g., with the event centred over the LT) would fit the observed debris flow patterns much better. In addition, detailed geomorphological mappings in the study area would further improve the understanding of the spatial differences in the debris flow distribution.

The LiDAR data also show a large variability of different debris flow magnitudes, with accumulation volumes up to 40,000 m$^3$. These magnitudes show a power–law relationship with the amount of rainfall within the respective debris flow catchments. Although the relationship reveals a large scatter of the possible accumulation volumes, it can be used to improve or validate debris flow prediction models.

This study shows that the combination of several data sets can provide valuable information about the spatial characteristics of a debris flow event. It thus contributes to a better understanding of the triggering mechanisms and magnitudes of such events, which is important for hazard assessment of future events.

**Author Contributions:** Conceptualization, J.R., F.H. (Florian Haas), T.H. and M.B.; methodology, J.R., F.H. (Florian Haas), F.F., M.A. and T.H.; formal analysis, J.R. and F.H. (Florentin Hofmeister); investigation, J.R., F.H. (Florian Haas), F.H. (Florentin Hofmeister) and M.P.; data curation, J.R., F.H. (Florentin Hofmeister) and M.P.; writing—original draft preparation, J.R. and F.H. (Florentin Hofmeister); writing—review and editing, J.R., F.H. (Florian Haas), F.H. (Florentin Hofmeister), F.F. and M.A.; visualization, J.R.; supervision, F.H. (Florian Haas), T.H. and M.B.; project administration, F.H. (Florian Haas), T.H. and M.B.; funding acquisition, F.H. (Florian Haas), T.H. and M.B. All authors have read and agreed to the published version of the manuscript.

**Funding:** This research has been funded by the Deutsche Forschungsgemeinschaft (DFG, German Research Foundation; project number 394200609 (FOR 2793)), the Austrian Science Fund, the autonomous province of South Tyrol and the Swiss National Science Foundation.

**Data Availability Statement:** The meteorological data from the stations Horlachalm and Niederthai, as well as the hydrological data from the stations Horlach Fassung and Stuibenfall, are provided by the Tyrolean Hydropower Company (TIWAG) and are not available due to commercial restrictions. INCA data are publicly available and can be found here: https://data.hub.zamg.ac.at/dataset/inca-v1-1h-1km (accessed on 13 February 2023). The meteorological data from Grastal station, the hydrological data from Finstertal and Larstigtal as well as the 2021 and 2022 LiDAR data will be publicly available after completion of the SEHAG research project and can be provided upon request.

**Acknowledgments:** We would like to thank TIWAG for the fast support with their data and the Zentralanstalt für Meteorologie und Geodynamik (ZAMG) for their help with the INCA dataset. Special thanks to all the student assistants who supported our field work as well as all residents of Niederthai and Umhausen for supporting the research projects in Horlachtal. In addition, we want to thank the two anonymous reviewers for their help to improve our manuscript.

**Conflicts of Interest:** The authors declare no conflict of interest.

### Appendix A

Figure A1 shows the 23 areas surrounding the debris flow process areas that served as subsets for further co-registration of the two LiDAR datasets from 2021 and 2022. For each of these areas, an ICP as well as a raster-based registration was applied to the pointclouds.

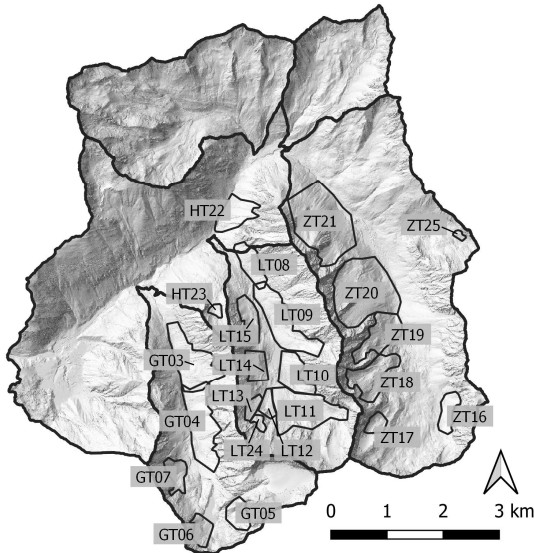

**Figure A1.** The location of the 23 subsets for pointcloud registration.

For the DoD in each of the subsets (see Figure A1), an error assessment was applied by using stable areas as a reference. Statistical measures of the respective calculations are shown in Table A1.

**Table A1.** Statistical measures of the DoD for the different subsets.

| Subset Area ID | Mean Error [m] | Mean Absolute Error [m] | RMSE [m] | Standard Deviation [m] |
|---|---|---|---|---|
| GT03 | −0.008 | 0.083 | 0.111 | 0.111 |
| GT04 | 0.005 | 0.061 | 0.08 | 0.079 |
| GT05 | 0.002 | 0.129 | 0.171 | 0.171 |
| GT06 | −0.036 | 0.196 | 0.257 | 0.255 |
| GT07 | −0.025 | 0.23 | 0.28 | 0.278 |
| LT08 | −0.008 | 0.072 | 0.097 | 0.096 |
| LT09 | 0.033 | 0.113 | 0.199 | 0.196 |
| LT10 | −0.0001 | 0.072 | 0.098 | 0.098 |
| LT11 | 0.013 | 0.154 | 0.197 | 0.197 |
| LT12 | −0.039 | 0.25 | 0.309 | 0.307 |
| LT13 | 0.01 | 0.26 | 0.323 | 0.323 |
| LT14 | 0.002 | 0.184 | 0.227 | 0.227 |
| LT15 | 0.018 | 0.249 | 0.307 | 0.306 |
| LT24 | −0.053 | 0.159 | 0.211 | 0.205 |
| ZT16 | −0.015 | 0.119 | 0.173 | 0.173 |
| ZT17 | −0.006 | 0.074 | 0.099 | 0.099 |
| ZT18 | −0.003 | 0.09 | 0.121 | 0.121 |
| ZT19 | 0.009 | 0.088 | 0.113 | 0.113 |
| ZT20 | −0.002 | 0.075 | 0.106 | 0.106 |
| ZT21 | −0.003 | 0.076 | 0.102 | 0.102 |
| ZT25 | −0.0001 | 0.044 | 0.055 | 0.055 |
| HT22 | 0.001 | 0.096 | 0.124 | 0.124 |
| HT23 | 0.001 | 0.132 | 0.169 | 0.169 |

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
