# Peer review of "Analysing the Large-Scale Debris Flow Event in July 2022 in Horlachtal, Austria Using Remote Sensing and Measurement Data"

_geosciences, doi:10.3390/geosciences13040100_

Round 1

Reviewer 1 Report

Dear authors,

thank you very much for your paper.

The paper titled "Analysing the large-scale debris flow event in July 2022 in Horlachtal, Austria using remote sensing and measurement data" offers many interesting clues about the feasibility of an approach based on high-resolution LiDAR, rainfall and hydrological data, and automated collection of photographic documentation for debris flows event analysis at the catchment scale within the Alpine context.

The article is found to be very clear in describing the used data and the applied methodology. The results are presented in a clear and effective way.

However, in my opinion, there are some issues that should be solved in the manuscript:

1.      The study area is described in a concise paragraph, with scarce reference to its geological and geomorphological features. In addition, the little information provided relates only to bedrock with no mention of continental deposit characteristics or the geomorphological setting and morphometric characteristics of the area. These last are widely recognized in the literature as key aspects for characterizing basins from a debris flow susceptibility perspective (e.g. Calista et al., 2020; Palamakumbura et al., 2021; Pasculli et al. 2021; Tiranti et al, 2018).

2.      Section 5.2 discusses the spatial distribution of debris flows that occurred during the July 2022 event. Given the above, this analysis does not consider the distribution of loose deposits, their thicknesses, and the spatial distribution of major morphometric parameters (e.g., slope, aspect, curvature). In my opinion, taking into account the above aspects as well could open up important implications for the discussion of the spatial distribution of events.

Please consider the other minor suggestions below:

1.      In Figure 1: Please consider adding labels to indicate subfigures, as done, for example, in Figure 2; in the location map of the study area in the Stubai Alps, please consider indicating some of the major cities in order to make easier understand where the study area is located.

2.      In figure 2: Please consider indicating photo orientation and providing a proper scale.

3.      In figure 8: The location of the shown levées within the study area should be indicated.

According to these suggestions, it is my opinion that the article in its present form requires minor revisions to be suitable for publication.

References:

a.        Calista, M., Menna, V., Mancinelli, V., Sciarra, N., & Miccadei, E. (2020). Rockfall and Debris Flow Hazard Assessment in the SW Escarpment of Montagna del Morrone Ridge (Abruzzo, Central Italy). Water, 12(4), 1206. MDPI AG. Retrieved from http://dx.doi.org/10.3390/w12041206.

b.       Palamakumbura, R., Finlayson, A., Ciurean, R., Nedumpallile-Vasu, N., Freeborough, K., & Dashwood, C. (2021). Geological and geomorphological influences on a recent debris flow event in the Ice-scoured Mountain Quaternary domain, western Scotland. Proceedings of the Geologists' Association, 132(4), 456-468. https://doi.org/10.1016/j.pgeola.2021.05.002.

c.        Pasculli, A., Cinosi, J., Turconi, L., & Sciarra, N. (2021). Learning Case Study of a Shallow-Water Model to Assess an Early-Warning System for Fast Alpine Muddy-Debris-Flow. Water, 13(6), 750. MDPI AG. Retrieved from http://dx.doi.org/10.3390/w13060750.

d.       Tiranti, D., Crema, S., Cavalli, M., & Deangeli, C. (2018). An integrated study to evaluate debris flow hazard in alpine environment. Frontiers in Earth Science, 6, 60. https://doi.org/10.3389/feart.2018.00060.

Reviewer 2 Report

Review of manuscript "Analysing the large-scale debris flow event in July 2022 in Horlachtal, Austria using remote sensing and measurement data" (geosciences-2254392)

Dear authors, your research aims to present quantitative analyses of an extreme debris flow event in the Horlachtal, Austria, triggered by local high-intensity short-duration precipitation events on 20th and 23rd of July 2022. It is a well-prepared manuscript and fits the aims and scope of the journal topic. Nevertheless, the authors need to highlight the soundness and novelty of their research as compared with previous research. Therefore, "Major Revision" is necessary to improve this manuscript. Specifically, the reviewer has the following comments and suggestions:

(1) The Introduction Section and Abstract: these two parts are not strong because the authors did not highlight the necessity and novelty of this study from an international perspective. As a consequence, reviewers cannot figure out why this research must be performed in this context. For example, precise data on debris flow triggering thresholds, accumulation volumes and spatial characteristics of large-scale events are scarce. But what are the reasons and how did you resolve it?

(2) The Literature part: the contents are weak in this part, and the authors need to look further into the relevant research about geohazard risk assessment, such as those using the maximum entropy method.

(3) Please explain why a small study area was selected in this study.

(4) The authors devote too much space to describing the background of the debris flow issues in the study area, which should be briefly mentioned in two or three simple subsections or paragraphs. The Introduction section is meant to set the context for your research work and highlight how it contributes to the knowledge in this field and builds on previous similar studies.

(5) Section 3. Materials and Methods: in this part, please provide all the necessary detailed information on the data used in this study, such as the acquiring dates, accuracies, preprocessing methods, and spatial resolution, in a new Table.

(6) In addition, a technical roadmap of this study should also be provided as a new figure.

(7) Section 3.3. Meteorological Data: in this part, did the authors obtain the final precipitation data by using interpolation methods?

(8) In the Discussion Sections, the relationship between this study and previous studies should also be mentioned and discussed. In particular, the advanced maximum entropy method has been largely used in geohazard susceptibility assessment (please find below). Nevertheless, this new method was ignored in this manuscript.

https://doi.org/10.1016/j.scs.2022.103812

https://doi.org/10.1016/j.jenvman.2019.02.020

https://doi.org/10.1038/s41598-021-85862-7

(9) The field investigation may not be enough because of the extremely large debris flow-prone areas.

(10) Please also improve the Conclusions by clarifying the main limitations of your work.

Round 2

Reviewer 2 Report

Both the organization and language usage need to be improved.